 

# Dysregulated Ca²⁺ signaling, fluid secretion, and mitochondrial function in a mouse model of early Sjögren's disease

Kai-Ting Huang[1], Larry E Wagner[1], Takahiro Takano[1], Xiao-Xuan Lin[1], Harini Bagavant[2], Umesh Deshmukh[2], David I Yule[1]*

[1]Department of Pharmacology and Physiology, University of Rochester, Rochester, United States; [2]Arthritis and Clinical Immunology, Oklahoma Medical Research Foundation, Oklahoma City, United States

*For correspondence:
david_yule@urmc.rochester.edu

**Competing interest:** The authors declare that no competing interests exist.

**Abstract** The molecular mechanisms leading to saliva secretion are largely established, but factors that underlie secretory hypofunction, specifically related to the autoimmune disease Sjögren's syndrome (SS) are not fully understood. A major conundrum is the lack of association between the severity of salivary gland immune cell infiltration and glandular hypofunction. SS-like disease was induced by treatment with DMXAA, a small molecule agonist of murine STING. We have previously shown that the extent of salivary secretion is correlated with the magnitude of intracellular Ca²⁺ signals (Takano et al., 2021). Contrary to our expectations, despite a significant reduction in fluid secretion, neural stimulation resulted in enhanced Ca²⁺ signals with altered spatiotemporal characteristics in vivo. Muscarinic stimulation resulted in reduced activation of the Ca²⁺-activated Cl⁻ channel, TMEM16a, although there were no changes in channel abundance or absolute sensitivity to Ca²⁺. Super-resolution microscopy revealed a disruption in the colocalization of Inositol 1,4,5-trisphosphate receptor Ca²⁺ release channels with TMEM16a, and channel activation was reduced when intracellular Ca²⁺ buffering was increased. These data indicate altered local peripheral coupling between the channels. Appropriate Ca²⁺ signaling is also pivotal for mitochondrial morphology and bioenergetics. Disrupted mitochondrial morphology and reduced oxygen consumption rate were observed in DMXAA-treated animals. In summary, early in SS disease, dysregulated Ca²⁺ signals lead to decreased fluid secretion and disrupted mitochondrial function contributing to salivary gland hypofunction.

## eLife assessment

This manuscript presents **important** observations on the early changes that occur in calcium signaling, TMEM16a channel activation, and mitochondrial dysfunction in salivary gland cells in a murine model of autoimmune Sjögren's disease. The study reports that in response to DMXAA treatment which induces a murine model of Sjögren's disease, salivary gland cells show significant changes in saliva release, calcium signaling, TMEM16a activation, mitochondrial function, and subcellular morphology of the endoplasmic reticulum. The work is **compelling** and will be of strong interest to physiologists working on secretion, calcium signaling, and mitochondria.

## Introduction

Saliva plays crucial roles in oral health, including lubricating the mouth, maintaining pH balance, defense against microorganisms, aiding taste, and initiating digestion of macronutrients (*Humphrey and Williamson, 2001*; *Carpenter, 2013*; *Pedersen et al., 2018*). Saliva is produced primarily by

three major salivary glands; the submandibular gland (SMG), parotid gland (PG), sublingual gland (SLG), and some minor glands in the lower lip, tongue, and cheek. Saliva is generated in secretory acinar cells, with its content adjusted by ducts before reaching the mouth. The acinar cells are fundamental to the production of the primary salivary secretion (*de Paula et al., 2017*). The fluid secretion process is driven by the trans-epithelial movement of $Cl^-$ across acinar cells. To accomplish vectorial movement of $Cl^-$, acinar cells are polarized such that the basolateral plasma membrane (PM) faces the interstitium and is adjacent to blood vessels, while the apical PM forms a lumen with the distinct PM regions physically segregated by tight-junctional complexes. At the basolateral PM, $Cl^-$ is transported into the acinar cell cytoplasm against their electrochemical gradient *via* the $Na^+/K^+/2Cl^-$ cotransporter, (NKCC1). Following mastication or the experience of the taste and the smell of food, the neurotransmitter, acetylcholine (ACh) is released from parasympathetic nerves and acts on muscarinic receptors on the basolateral PM. Activated muscarinic receptors promote the production of inositol 1,4,5 trisphosphate ($IP_3$), and subsequently $Ca^{2+}$ release from endoplasmic reticulum (ER) stores *via* $IP_3$ receptors ($IP_3Rs$) situated in the ER in the extreme apical aspects of the cell (*Futatsugi et al., 2005*; *Lee et al., 1997*; *Pages et al., 2019*). Elevated $[Ca^{2+}]_i$ activates a $Ca^{2+}$-activated $Cl^-$ channel named TMEM16a that allows $Cl^-$ to move through the apical PM to the ductal lumen which is continuous with the salivary intercalated duct (*Pedersen et al., 2018*; *Melvin et al., 2005*). In turn, $Na^+$ moves through the paracellular space to balance the $Cl^-$ and water follows osmotically both paracellularly and through the water channel aquaporin5 (AQP5) to generate the primary saliva (*Melvin et al., 2005*).

The importance of saliva is underappreciated in the absence of hypofunction. Reduced salivary secretion is termed xerostomia and can result from the iatrogenic effects of drugs, as collateral damage to salivary glands following radiotherapy for malignancy in the head and neck area, and commonly in SS (*Saleh et al., 2015*). SS is a chronic autoimmune disorder, that is predominantly manifested as profound dry eye and dry mouth as ultimately the immune system targets and destroys lacrimal and salivary gland cells (*Tabbara and Vera-Cristo, 2000*; *Ramos-Casals et al., 2012*; *Mavragani and Basiaga, 2014*; *Brito-Zerón et al., 2016*). SS can occur independently (primary SS, pSS) or concurrently with diseases such as arthritis or lupus (secondary SS, sSS) (*Kiripolsky et al., 2017*). SS affects millions of people, predominantly females in their fourth and fifth decades of life. While treatments can alleviate symptoms, there is no cure or intervention to halt its progression. The etiology of SS remains largely unresolved, but it's believed to result from a combination of genetic, environmental, hormonal, and possibly viral factors, causing an aberrant immune response directed against the exocrine glands. The identification of SS usually is scored by the extent of salivary hypofunction, the degree of immune infiltration, evidence of damage to minor salivary glands observed following biopsy, and the presence of autoantibodies, such as anti-SSA (Ro) and Anti-SSB (La) and anti-nuclear antibody (ANA) which are classically found in SS (*Mavragani and Basiaga, 2014*; *Brito-Zerón et al., 2016*; *Jonsson et al., 2018*). Notably, however, in the early phases of SS, there is minor immune infiltration and little overt damage to exocrine tissue despite profound hypofunction. Provocatively, these data indicate that loss of secretory tissue per se is not the causative event resulting in dryness early in the disease, and further indicates that a defect in the stimulus-secretion coupling mechanism precedes glandular destruction and possibly contributes to the progression of the disease.

Over the years, numerous mouse models both genetic and 'induced' have been developed to study the pathogenesis of SS, with each exhibiting specific aspects of the human condition, including glandular dysfunction, autoantibody production, and lymphocytic infiltration (*Lee et al., 2012*; *Gao et al., 2020*). To investigate the early events in SS, we concentrated on an SS model induced by the activation of the stimulator of the interferon gene (STING) pathway. This is thought to mirror the molecular response to bacterial/viral infection. STING is primarily located in the ER and plays a crucial role in the innate immune response, especially against DNA viruses and intracellular bacteria. Activation of STING occurs upon sensing cytosolic DNA as a result of cell damage or from microbial origin following infection. When cytosolic DNA is detected, it is first recognized by a sensor molecule called cGAS (cyclic GMP-AMP synthase). Binding to DNA prompts cGAS to generate cGAMP (cyclic GMP-AMP), which, in turn, binds to and activates STING (*Decout et al., 2021*). Once STING is activated, it undergoes a series of transformations that ultimately result in the transcription of type I interferon genes, especially the gene encoding interferon-β (IFN-β) (*Papinska et al., 2018*). The production of type I interferons is a primary antiviral response and a significant characteristic of SS (*Papinska et al., 2018*; *Huijser et al., 2022*). STING can be activated pharmacologically by exposure to 5,6-Dimethyl-9

-oxo-9*H*-xanthene-4-acetic acid (DMXAA), which faithfully reproduces the immune response observed following physiological STING activation (*Gao et al., 2013*; *Weiss et al., 2017*; *Cerón et al., 2019*).

In this study, we investigated the early events in the initiation of SS-like disease that lead to salivary gland hypo-function using the DMXAA SS model. We first utilized in vivo intravital imaging to investigate any potential dysregulation of $Ca^{2+}$ signaling in the DMXAA-induced SS mouse model. Paradoxically, the spatially averaged $Ca^{2+}$ levels achieved following neural stimulation in mice treated with DMXAA were enhanced despite significantly reduced fluid secretion. Notably, however, the stereotypical spatial characteristics of the $Ca^{2+}$ signal were disrupted. Downstream of the $Ca^{2+}$ signal, the activity of the TMEM16a $Ca^{2+}$-activated Cl channel stimulated by muscarinic secretagogues was reduced, despite no changes in the abundance or localization of the protein or absolute sensitivity to activation by $Ca^{2+}$. The intimate localization of $IP_3R$ and TMEM16a was, however, disrupted, contributing to the reduced activity of TMEM16a upon agonist stimulation as local peripheral coupling between channels is disrupted. Moreover, we observed disordered mitochondrial morphology, abundance, and function in the disease model. These data suggest that early in SS, reduced fluid secretion occurs because of a defect in the secretagogue activation of Cl⁻ secretion. Further significant mitochondrial dysfunction is evident, possibly as a result of the aberrant $Ca^{2+}$ signals that may contribute to the dysregulated $Ca^{2+}$ signals and/or progression of SS disease.

## Results

### Saliva secretion is attenuated in both SMG and PG in the SS mouse model

Activation of the STING pathway in mice has been established as a model for the initiation of SS. This pathway is normally activated following exposure to foreign nucleic acids and is thought to mimic exposure of cells to DNA/RNA from viruses and bacteria. Activation of this pathway in salivary glands is characterized by initiation of a type-1 interferon response, mild immune cell infiltration, and a marked loss of saliva secretion without obvious morphological damage and, therefore, mimics the early clinical manifestations of SS disease. Thus, to investigate the early cellular events in acinar cells during the initiation of SS in mice, we chose to pharmacologically activate this pathway using DMXAA, a STING pathway agonist. As described in Methods, DMXAA (or control solution) was administered on day 0 and day 21 of the experimental timeline (*Figure 1A*). Immunofluorescent staining in sliced SMG tissue indicated that STING protein was increased in SMG in the DMXAA-treated mouse on day 28, seven days after the final DMXAA administration (*Figure 1—figure supplement 1*) confirming the activation of the STING pathway. Whole saliva production from the major salivary glands was evaluated on day 28 following systemic stimulation with the muscarinic receptor agonist, pilocarpine. To avoid potential weight-related variations in saliva secretion, the total saliva output was normalized to the individual mouse's body weight. Notably, the average saliva production was reduced from 130.1±48.96 mg in vehicle-treated mice to 63.71±30.41 mg in DMXAA-treated mice, a reduction in saliva production of 48.97% (*Figure 1B*). Consequently, DMXAA treatment resulted in 51.99% saliva production compared to vehicle-treated mice (*Figure 1C*). Moreover, H&E staining indicated that mild immune infiltration was observed in the DMXAA-treated mice with no overt changes to the morphology of the gland (*Figure 1D*). The mass of the SMG was also not significantly different in DMXAA vs. vehicle-treated animals, consistent with no loss of secretory tissue (*Figure 1E*). Collectively, these results suggest that DMXAA-treated mice exhibit characteristics of early-stage SS and could be a useful model for investigating the pathophysiological mechanisms underlying secretory dysfunction and advancing our understanding of the disease's progression.

To further investigate the individual relative contribution of the SMG and PG to the decrease in total saliva secretion using a more physiological stimulation paradigm, we performed experiments where the nerve bundle innervating a particular gland was electrically stimulated and saliva secretion quantitated. Previous research in our lab has established the range and parameters for physiological stimulation of secretion (*Takano et al., 2021*). The production of saliva was significantly diminished in DMXAA-treated animals compared with vehicle controls at stimulation frequencies of 7 and 10 Hz in SMG (*Figure 1F*) and at 5, 7, and 10 Hz in the PG (*Figure 1G*). These findings confirm that activation of the STING pathway reduces the function of both SMG and PG, consistent with the diminished

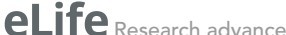

**Figure 1.** Deficiency in salivary secretion in 5,6-Dimethyl-9-oxo-9*H*-xanthene-4-acetic acid (DMXAA)-induced Sjögren's syndrome (SS) mouse model.
(**A**) Schematic timeline for the generation of the SS mouse model. Female wild-type (WT) mice were administered two subcutaneous doses of DMXAA on Day 0 and Day 21. Salivary gland function was assessed on day 28. (**B–C**) Saliva, stimulated by pilocarpine, was collected over 15 min. (**B**) The amount of saliva secretion was determined by measuring the saliva weight. Vehicle: n=30 mice, SS mouse model: n=32 mice. Mean ± SD. (**C**) The weight of collected saliva was normalized to each mouse's body weight. Vehicle: n=26 mice, SS mouse model: n=29 mice. Mean ± SD. Unpaired two-tailed t-test.
(**D**) H&E stained sections from the vehicle or DMXAA-treated animals. Treated animals showed minor lymphocyte infiltration and inflammation as focal peri-vascular/peri-ductal lymphocytic sialoadenitis adjacent to normal-looking acini. (**E**) The glandular damage was assessed by normalizing the weight of the submandibular gland (SMG) to the mouse's body weight. Each dot represents the weight of one SMG. n=10 from 5 mice for both vehicle-treated and DMXAA-treated mice. (**F–G**) A comparison of total saliva secretion following 1 min stimulations at the indicated frequency (**F**) from the SMG of mice

*Figure 1 continued on next page*

*Figure 1 continued*

(Vehicle: n=8 mice, SS mouse model: n=6) and (**G**) from the parotid gland (PG) (Vehicle: n=7 mice, SS mouse model: n=5 mice). Mean ± SD. Two-way ANOVA with multiple comparisons. Source data is included in *Figure 1—source data 1*.

The online version of this article includes the following source data and figure supplement(s) for figure 1:

**Source data 1.** Raw data of amounts of salivary secretion for individual animals.

**Figure supplement 1.** Up-regulation of stimulator of the interferon gene (STING) protein expression in both submandibular gland (SMG) and parotid gland (PG) treated with Dimethyl-9-oxo-9H-xanthene-4-acetic acid (DMXAA).

**Figure supplement 1—source data 1.** Raw data for the amount of STING fluorescence per cell.

production of whole saliva. Notably, the reduction in function of PG, the gland responsible for the majority of stimulated saliva secretion, was relatively greater than in SMG.

## Altered spatiotemporal characteristics of Ca²⁺ signals in the SS mouse model

An increase in intracellular $Ca^{2+}$ plays a central role in regulating the cellular machinery underlying the fluid secretion mechanism. In particular, as noted, an increase in $Ca^{2+}$ is important for the activation of ion channels localized in particular domains of the polarized acinar cell which play a central role in saliva secretion (*Takano et al., 2021*). Given that the precise spatiotemporal characteristics of the $Ca^{2+}$ signal in salivary acinar cells are thought to be fundamental to the appropriate activation of the fluid secretion machinery, we evaluated whether hypofunction following DMXAA treatment resulted from dysregulation of the stimulated $Ca^{2+}$ signal. Previous research in our lab developed a platform to study $Ca^{2+}$ signaling in vivo using Multiphoton (MP) imaging in transgenic mice engineered to express a genetic-encoded $Ca^{2+}$ indicator, GCaMP6f, specifically in acinar cells (*Takano et al., 2021*). These mice were generated by crossing homozygous mice expressing the fast genetically encoded $Ca^{2+}$ indicator GCaMP6f (B6J.Cg-*Gt(ROSA)26Sor*$^{tm95.1(CAG-GCaMP6f)Hze}$/MwarJ) floxed with a STOP cassette with heterozygous tamoxifen-inducible Mist1 Cre mice (B6.129-*Bhlha15*$^{tm3(cre/ERT2)Skz}$/J). The protocol for STING pathway induction was applied to the mice expressing GCamp6f (*Figure 2—figure supplement 1B*). Salivary gland function was assessed by the amount of pilocarpine-induced saliva secretion. The secretion deficiency observed in wild-type mice was recapitulated in these mice from a different genetic background (*Figure 2—figure supplement 1B*). We reasoned that decreased fluid secretion could result from reduced or dysregulated $Ca^{2+}$ signaling in DMXAA-treated mice. Therefore, next, we compared the spatially averaged $Ca^{2+}$ signal evoked by direct nerve stimulation in DMXAA-treated vs. vehicle control animals in vivo in SMG. SMG were stimulated at frequencies optimum for fluid secretion (1–10 Hz) for 10 s and the $Ca^{2+}$ signals were recorded. *Figure 2A* shows standard deviation (SD)-projection images visualizing the spatial and amplitude changes in $Ca^{2+}$ throughout the field of view during the entire period of stimulation at the indicated frequencies. In vehicle-treated mice, the $Ca^{2+}$ signals at low stimulus strengths occurred in a minority of the cells and predominantly propagated below the apical PM. As the stimulation strength increased, $Ca^{2+}$ signals became more pronounced, and more acinar cells responded. Strikingly, acinar cells in DMXAA-treated mice demonstrated enhanced sensitivity to stimulation. At lower stimulation strengths, a larger number of acinar cells responded, and the spatially averaged $Ca^{2+}$ signals in these cells were notably larger when compared to those in the control group. *Figure 2B* shows a time series of images following 7 Hz stimulation. This augmented response was manifested as an elevated maximum peak [$Ca^{2+}$] (*Figure 2D*), shorter latency (*Figure 2E*), and larger area under the curve (AUC) during stimulation in DMXAA-treated animals (*Figure 2F*).

In addition to the absolute magnitude of the spatially averaged $Ca^{2+}$ signal, the subcellular spatial characteristics of the stimulated $Ca^{2+}$ rise are also important for appropriate stimulation of fluid secretion (*Takano et al., 2021*). $Ca^{2+}$ signals stimulated following nervous stimulation in SMG are invariably initiated in the extreme apical pole of acinar cells and subsequently establish a standing gradient that dissipates rapidly to result in apically confined signals that do not substantially propagate to the basal aspects of the cell following physiological stimulation (*Takano et al., 2021*). We, therefore, investigated if the spatial characteristics of stimulated $Ca^{2+}$ signals were altered in DMXAA-treated animals. SD image projections generated during the period of stimulation demonstrated that the [$Ca^{2+}$]ᵢ increase was tightly localized below the apical PM within the acinar cells in the vehicle-treated animals

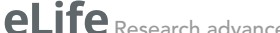

**Figure 2.** Augmented global Ca²⁺ signals in vivo in Sjögren's syndrome (SS) mouse model. (**A**) Representative standard deviation images of Ca²⁺ signals during the 10 s of stimulation. Scale bar: 26 µm (**B**) The time-course of pseudo-color images of Ca²⁺ in response to 7 Hz stimulation. Scale bar: 26 µm (**C**) Representative cellular responses to stimulation at the indicated frequencies averaged from the entire cell. n=10 cells, one animal. (**D**) A comparison of peak Ca²⁺, (**E**) area under a curve, and (**F**) latency during each stimulation in submandibular gland (SMG). Each symbol represented the average

*Figure 2 continued on next page*

*Figure 2 continued*

response of ten cells from one view. Vehicle: n=3–6 from three mice; SS mouse model: n=8–10 from four mice. Mean ± SD. Two-way ANOVA with multiple comparisons. Source data is included in .

The online version of this article includes the following source data and figure supplement(s) for figure 2:

**Source data 1.** Raw data for salivary secretion from individual animals.

**Figure supplement 1.** Deficiency in secretion in transgenic animals expressing GCamp6f following Dimethyl-9-oxo-9H-xanthene-4-acetic acid (DMXAA) treatment.

**Figure supplement 1—source data 1.** Raw data for the amount of saliva secretion for individual animals.

(*Figure 3A*). However, in the DMXAA-treated animals, the $[Ca^{2+}]_i$ exhibited a more global distribution through the entire cell cytoplasm (*Figure 3B*). The $[Ca^{2+}]_i$ was visualized *via* line-scan plots revealing the temporal alterations along a line extending from the apical PM to the basolateral PM, traversing the nucleus, over time within an acinar cell. A significant $[Ca^{2+}]_i$ elevation was evident at the basolateral aspects of the acinar cell in the DMXAA-treated animals (*Figure 3D*) when compared to the vehicle-treated control (*Figure 3C*). As shown in the kinetic plots, the $[Ca^{2+}]_i$ in the apical region is greater in DMXAA-treated mice compared to the vehicle-treated mice (*Figure 3E*). Moreover, a significant $Ca^{2+}$ signal was observed in the extreme basal region of the cell in DMXAA but not in vehicle-treated animals (*Figure 3F*). The comparison of $Ca^{2+}$ signal ratios at the apical versus basolateral PM indicated the most significant globalization of the $Ca^{2+}$ signal occurred at 10 Hz stimulation (*Figure 3G*) which corresponds to the stimulation strength that results in maximal fluid secretion (*Takano et al., 2021*). In total, these data demonstrate that the magnitude of the spatially averaged $Ca^{2+}$ signal, together with the spatiotemporal characteristics of the signal are altered in the DMXAA-treated animals, but that these changes can not readily account for the reduction in fluid secretion observed.

## Secretagogue-stimulated TMEM16a activity is suppressed in the SS mouse model

The rate-limiting step for the secretion of fluid is the activation of the $Ca^{2+}$-activated $Cl^-$ channel, TMEM16a. We considered that a reduction in fluid secretion could conceptually occur by a reduction or mislocalization of TMEM16a protein, or by compromised muscarinic receptor-stimulated activation of the channel. Western blotting indicated the TMEM16a protein expression was comparable between the vehicle and SS mouse models (*Figure 4A and B*). In addition, immunolocalization using confocal microscopy demonstrated that TMEM16a localization remained largely unchanged in the SS mouse model (*Figure 4C*). Thus, a decrease in protein expression or mislocalization of the protein does not result in a reduction in stimulated saliva secretion. Similarly, the expression and localization of Aquaporin 5 were also not altered (*Figure 4—figure supplement 1A–C*). We next investigated whether the activation of this ion channel was compromised in DMXAA-treated animals using whole-cell patch clamp electrophysiology. In the absence of stimulation, no $Cl^-$ currents were observed in either vehicle or DMXAA animals following either depolarizing or hyperpolarizing voltage steps from a holding potential of –50 mV (*Figure 4D*). In the presence of 1 μM of the muscarinic agonist Carbachol (CCh), robust $Cl^-$ currents were measured in acini prepared from vehicle-treated animals (*Figure 4E*), which were greatly reduced in DMXAA-treated animals (*Figure 4D and F*). Reduced CCh-stimulated $Cl^-$ currents could potentially occur because of altered $Ca^{2+}$ regulation of TMEM16a following disruption of the spatial characteristics of the stimulated $Ca^{2+}$ signal. Theoretically, it is also possible that the $[Ca^{2+}]_i$ in the immediate vicinity of TMEM16a was disrupted, despite the augmented spatially averaged peak response. We, therefore, next tested whether TMEM16a activity stimulated directly by 0.5, 1 or, 5 μM $Ca^{2+}$ in the pipette solution (and thus globally in the cytoplasm) was altered in DMXAA-treated animals. Surprisingly, TMEM16a was activated to a similar extent by $Ca^{2+}$ in the SS mouse model (*Figure 5A and B*). In total, our data suggest that TMEM16a abundance, localization, or activity per se are not altered and thus do not explain the significant reduction in saliva secretion in the DMXAA-treated model.

An alternative mechanism could be that the microdomain between the apical ER $Ca^{2+}$ release sites and the apical PM TMEM16a is disrupted in the disease model, resulting in compromised local coupling between the ER and PM channels. Therefore, we investigated how the activation of TMEM16a was affected by buffering the CCh-stimulated cytosolic $Ca^{2+}$ with slow and fast $Ca^{2+}$ chelators. EGTA is a

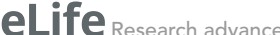

**Figure 3.** Disrupted spatial localization of Ca²⁺ signals in vivo in Sjögren's syndrome (SS) mouse model. (**A–B**) A representative standard deviation image during the 7 Hz stimulation in (**A**) vehicle condition and (**B**) in the SS mouse model. Scale bar: 26 μm. An acinus is outlined by the white broken line and a line from apical to basal is shown in red in each standard deviation (SD) image. (**C–D**) A representative 'kymograph' image of consecutive lines stacked in space over time for 7 Hz stimulation in (**C**) vehicle condition and (**D**) SS mouse model. Time is encoded along the X-axis from left to right. Space is encoded along the Y-axis from the apical side (bottom) to the basolateral side (top). Scale bar: 3 μm. (**E**) Representative trace of Ca²⁺ signals at 7 Hz nerve stimulation in an apical ROI generated as the initial 2 mm of the scanned line over time (yellow line) in vehicle-treated (black) and DMXAA-treated (orange) mice. The changes in apical ROI fluorescence at the indicated frequencies were quantified as the maximal Ca²⁺ changes normalized

*Figure 3 continued on next page*

*Figure 3 continued*

to the basal intensity. (**F**) Representative trace of $Ca^{2+}$ signals following 7 Hz nerve stimulation in a basolateral ROI generated as the final 2 mm of the scanned line (yellow line) over time in vehicle-treated (black) and DMXAA-treated (orange) mice. The changes in basolateral $Ca^{2+}$ signals at the indicated frequencies were quantified by the maximal $Ca^{2+}$ changes normalized to the basal intensity. (**G**) The ratio of the magnitude of $Ca^{2+}$ signal on the apical vs. the basolateral ROI upon stimulation at the indicated frequencies. Vehicle: n=5–6 replicates from three mice; SS mouse model: n=5–8 replicates from four mice. Mean ± SD. Two-way ANOVA with multiple comparisons. Source data is included in *Figure 3—source data 1*.

The online version of this article includes the following source data for figure 3:

**Source data 1.** Raw data for individual animals.

high-affinity $Ca^{2+}$ buffer with slow kinetics, while BAPTA has much more rapid kinetics. Experimentally, BAPTA can quickly buffer $Ca^{2+}$ changes close to $Ca^{2+}$ release sites and limit local activation of effectors within 20 nm. In contrast, EGTA has been shown to attenuate the rise in the bulk cytosol but is too slow to buffer local $Ca^{2+}$ in a restricted microdomain (*Eisner et al., 2023*). In the presence of BAPTA, no $Cl^-$ currents were detected in either vehicle-treated or DMXAA-treated animals (*Figure 6A*). However, in the presence of the slow $Ca^{2+}$ chelator, EGTA, CCH-stimulated $Cl^-$ currents were observed in acinar cells from vehicle-treated animals but were absent in the DMXAA-treated animals (*Figure 6A and B*). These data suggest that physiologically TMEM16a is activated by local changes in $Ca^{2+}$ signaling as a result of $IP_3$-induced $Ca^{2+}$ release in the apical domain of the acinar cell.

We noted that the abundance of $IP_3R2/3$ was not significantly altered in DMXAA-treated animals (*Figure 7—figure supplement 1A–C*), nevertheless next, we employed STED super-resolution microscopy to closely examine the spatial relationship between apical PM TMEM16a and $IP_3R3$ on the apical ER (*Figure 7A*). Despite the cell-cell contact distance remaining consistent in the disease model, as indicated by the distance between TMEM16a on the PM of adjacent acinar cells (*Figure 7F*), a notable increase in distance between the apical TMEM16a and $IP_3R3$ expressed on apical ER compared to the control group was observed (*Figure 7E*). In the control mice, the distance between TMEM16a and $IP_3R3$ was on average 84±17 nm, versus 155±20 nm in the SS disease mice. Similarly, the distance between $IP_3R3$ in adjacent cells was increased from 505±34 nm to 689±68 nm (*Figure 7C and D*). In total, these observations support the conclusion that the reduced activity of the TMEM16a channel is attributable to the disruption of the microdomain between TMEM16a and $IP_3R3$, such that the $Ca^{2+}$ flux through the $IP_3R$ is not communicated appropriately to its effector, TMEM16a.

## Compromised mitochondrial morphology and metabolism in the SS mouse model

$Ca^{2+}$ modulates cellular metabolism by the intricate bidirectional interaction between the ER and mitochondria. $Ca^{2+}$ transfer between ER and mitochondria is essential for optimal bioenergetics, and dysregulated $[Ca^{2+}]_i$ can be deleterious to mitochondrial function and alter morphology (*Duchen, 2000*; *Csordás et al., 2006*; *Ye et al., 2021*; *Katona et al., 2022*). The transfer of $Ca^{2+}$ between ER and mitochondria is dependent on the intimate physical localization of the organelles (*Katona et al., 2022*). Notably, aberrant mitochondrial morphology has been reported in the salivary glands of SS patients (*Barrera et al., 2021*). We first investigated mitochondrial abundance and morphology by immunofluorescence staining with antibodies directed against ATP5A, a component of the ATP synthesis machinery to visualize mitochondria, and $Na^+/K^+$ ATPase to localize the plasma membrane (*Figure 8A*). Using previously published methodologies (*Valente et al., 2017*; *Harwig et al., 2018*), quantification revealed a 22.16%±4.95 reduction in mitochondrial numbers in the SS mouse model relative to the vehicle-treated control (*Figure 8B*). Consistent with reduced mitochondrial numbers, less area was occupied by mitochondria in DMXAA-treated acinar cells (*Figure 8C*). Mitochondria morphology is intricately linked to their bioenergetic status We next evaluated mitochondrial morphology by their 'so-called' aspect ratio (AR) and form factor (FF) in DMXAA and vehicle-treated animals.The AR, the length of the major over minor axes of mitochondria documents the degree of fragmentation or elongation of individual mitochondria (*Katona et al., 2022*; *Harwig et al., 2018*). Mitochondria exhibited an 18.35%±4.62 decrease in mitochondrial elongation (*Figure 8D*) and a 20.7%±7.78 decrease in mitochondrial branching (*Figure 7E*) in the disease model compared to the vehicle-treated control condition. Importantly, these changes in mitochondrial number and morphology were not exclusive to the SMG as similar patterns were observed in the PG mitochondria,

**Figure 4.** Attenuated whole-cell macroscopic Cl⁻ currents induced by Carbachol (CCh) stimulation in Sjögren's syndrome (SS) mouse model. (**A**) Western blotting showing the protein expression level of TMEM16a in the vehicle condition and the 5,6-Dimethyl-9-oxo-9H-xanthene-4-acetic acid (DMXAA)-treated SS mouse model. Actin is the internal control. (**B**) The quantification of TMEM16a protein expression normalized to the internal control, Actin. Vehicle, n=4 mice; SS mouse model: n=6 mice. (**C**) Immunofluorescent staining in submandibular gland (SMG) tissue for TMEM16a (green), Na⁺/K⁺ ATPase (red), and DAPI for nucleus (blue). The upper panel is from the vehicle-treated control and the bottom panel is from DMXAA-

*Figure 4 continued on next page*

*Figure 4 continued*

treated animals. Scale bar: 30 μm. Unpaired two-tailed t-test. (**D**) Cl- currents when cells were held at –50 mV and stepped from –80–120 mV in 20 mV increments. (**E**) Time-dependent Cl- current density changes in response to the Carbachol (CCh) in the isolated acinar cells in vehicle conditions and SS mouse model. (**F**) Current-voltage relationships were measured before and after the addition of CCh in vehicle conditions (n=three mice, 3–4 cells per mouse) and SS mouse model (n=three mice, 3–4 cells per mouse). TMEM16a currents in the treated mice were markedly reduced compared to the control mice. Black dots represent the vehicle-treated cells and orange squares represent DMXAA-treated cells. The open symbols represent no stimulation; the solid symbols represent CCh stimulation. Source data is included in ***Figure 4—source data 1 and 2***.

The online version of this article includes the following source data and figure supplement(s) for figure 4:

**Source data 1.** Raw densitometry and current magnitude data.

**Source data 2.** Original immunoblots.

**Figure supplement 1.** Aquaporin5 (AQP5), the water channel, remained the comparable expression and proper localization in submandibular gland (SMG) in the disease mouse model.

**Figure supplement 1—source data 1.** Raw densitometry data.

**Figure supplement 1—source data 2.** Original immunoblots.

again marked by reduced mitochondrial count, increased fragmentation, and decreased branching (***Figure 8—figure supplement 1A–E***).

Next, we utilized electron microscopy (EM) to investigate mitochondrial ultrastructure. At low magnification, acinar cells from control mice contained defined mitochondria and well-formed ER stacks (***Figure 9A***. blue arrow). In contrast, the ER structure was disrupted in the SS disease model (***Figure 9A′***). At higher magnification,the coordinated ER structure was largely absent in diseased mice (***Figure 9B and B′***), and the proximity between ER and mitochondria was disrupted (***Figure 9H and I***). Moreover, we also observed scattered mitochondrial cristae at the highest magnification (***Figure 9C′ , and G***). Consistent with immunofluorescence studies, quantification of EM micrographs revealed that mitochondria were smaller, more fragmented (***Figure 9D and E***), and rounder (***Figure 9F***) in shape. In summary, our results collectively indicate significant morphological alterations in mitochondria in the SS disease model.

Mitochondrial morphology is a dynamic process that is intimately associated with mitochondrial bioenergetics and alterations in both, occur in response to changes in cellular status (***Hogan et al., 2005***; ***Martínez et al., 2020***; ***Navaratnarajah et al., 2021***; ***Schaefer et al., 2019***; ***Zorova et al., 2018***) Therefore, we investigated if changes in morphology might be associated with the disrupted function of mitochondria in the disease model. We measured mitochondrial membrane potential ($\Delta \Psi_m$), established by the electrochemical H⁺ gradient, which is the driving force of ATP production. Isolated SMG

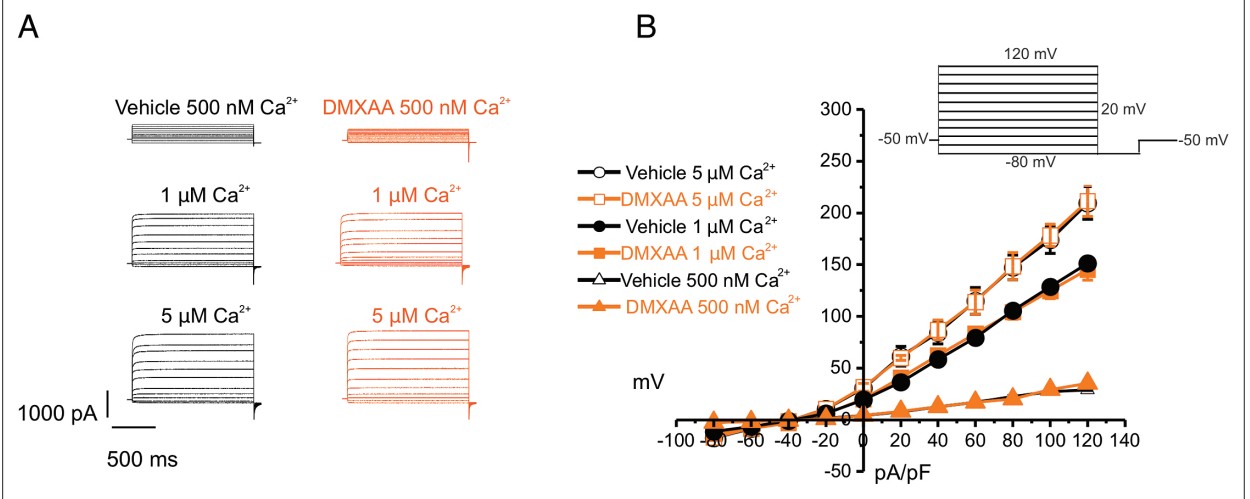

**Figure 5.** Increased $[Ca^{2+}]_i$ is capable of restoring TMEM16a functionality to 5,6-Dimethyl-9-oxo-9H-xanthene-4-acetic acid (DMXAA)-treated mice. (**A**) Cl- currents when cells were held at –50 mV and stepped from –80–120 mV in 20 mV increments. Either 0.5, 1, or 5 μM $[Ca^{2+}]_i$ in the patch pipette elicited a similar magnitude of Cl- currents for both the treated (n=3 mice, 3–4 cells per mouse) and control mice (n=3 mice, 3–4 cells per mouse). (**B**) Current-voltage relationships for both populations were essentially identical. Vehicle and SS mouse model: n=3 mice, 3–4 cells per mouse.

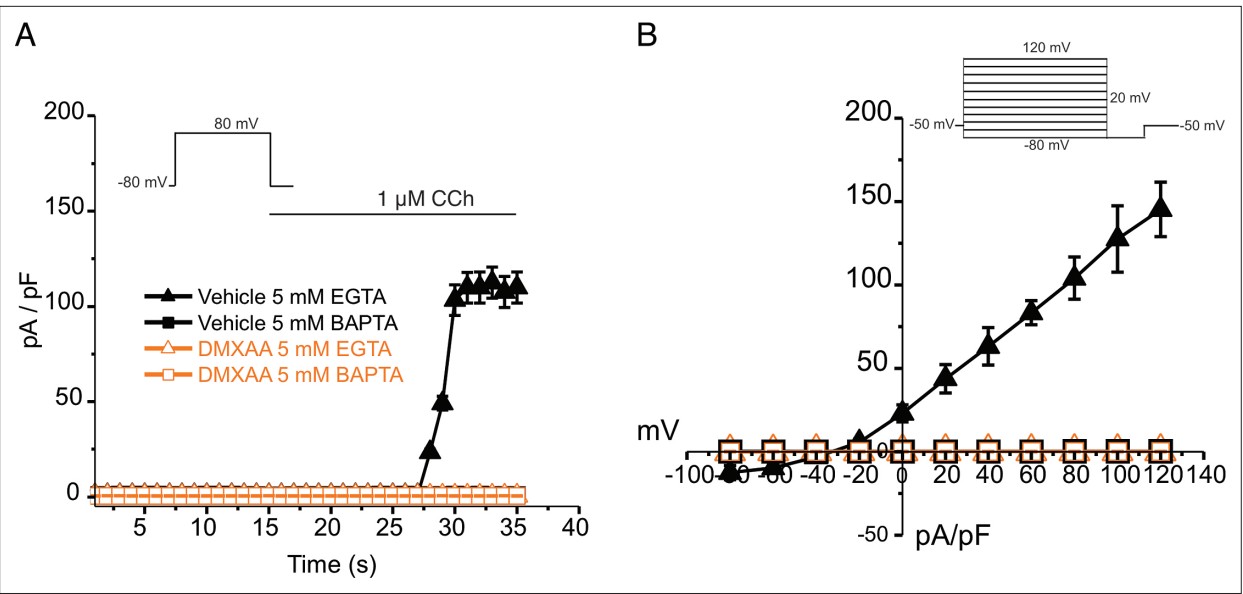

**Figure 6.** EGTA abolishes TMEM16a currents in 5,6-Dimethyl-9-oxo-9H-xanthene-4-acetic acid (DMXAA)-treated mice. (**A**) Cl⁻ currents in cells held at –80 mV and stepped to 80 mV with Carbachol (CCh) addition in EGTA (slow) and BAPTA (fast) buffered cells, respectively. (**B**) Current-voltage relationships were measured after the addition of CCh in 5 mM EGTA and 5 m M BAPTA-loaded isolated acinar cells from vehicle conditions (N=3 mice, 3–4 cells per mouse) and Sjögren's syndrome (SS) mouse model (n=3 mice, 3–4 cells per mouse). No TMEM16a currents in acini in either vehicle or DMXAA-treated mice in cells buffered with BAPTA. Triangles represent the 5 mM EGTA condition; squares represent the 5 mM BAPTA condition. The solid black symbols represent the vehicle-treated cells and hollow orange symbols represent DMXAA-treated cells.

acinar cells were loaded with TMRE, a $\Delta\Psi_m$-specific dye, and MitoTracker Green, to confirm mitochondrial localization and to facilitate the normalization of indicator loading. The maximal z-stacks projection images taken by confocal microscopy revealed colocalization of TMRE with MitoTracker Green (*Figure 10A*). The basal TMRE fluorescence was reduced in cells from DMXAA vs. vehicle-treated animals (*Figure 10—figure supplement 1*). To assess $\Delta\Psi_m$, we quantified the relative maximum dissipation of $\Delta\Psi_m$ in DMXAA and vehicle-treated acinar cellsby the mitochondrial uncoupler, FCCP (*Figure 10B*). Consistent with the reduction in basal TMRE fluorescence, the change in TMRE fluorescence normalized to mitochondrial content revealed a marked reduction in $\Delta\Psi_m$ in the acinar cells from the SS disease model (*Figure 10C*).

An appropriate $\Delta\Psi_m$ mitochondrial membrane potential is vital for maintaining bioenergetics (*Zorova et al., 2018*). Given that mitochondrial $\Delta\Psi_m$ was significantly depolarized in DMXAA-treated animals, we next evaluated the OCR, a key metric of mitochondrial bioenergetic function in isolated SMG acinar cells. We employed sequential exposure to agents that target the function of the mitochondrial electron transport chain (ETC) using Seahorse technology (*Figure 10D*). Our results revealed a 25% reduction in basal OCR in the SS model compared to the control animals (at –25.25±7.89 pmol/min; *Figure 10E*). While ATP-linked respiration showed no significant difference in post-oligomycin-induced ETC Complex V blockade in both conditions (*Figure 10F*). Intriguingly, the FCCP-provoked maximal respiration rate, an indicator of stress tolerance, remarkably declined by 47%±9.19 after FCCP treatment in the SS model (*Figure 10G*). These data indicate impaired mitochondrial function and stress responses in the SS mouse model.

## Discussion

SS is a complex inflammatory disease resulting from the intersection of genetics and environmental factors. This autoimmune disorder affects exocrine glands including salivary and lacrimal glands, leading to dry mouth and dry eyes, among other symptoms (*Ramos-Casals et al., 2012*; *Mavragani and Basiaga, 2014*; *Brito-Zerón et al., 2016*). SS animal models are crucial for understanding the pathogenesis, progression, and potential treatments for the disease, though like many animal models of disease, none can recapitulate all the aspects of SS. Currently, SS animal models are categorized

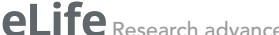

**Figure 7.** Disrupted proximity between TMEM16a and IP$_3$R3 in the 5,6-Dimethyl-9-oxo-9H-xanthene-4-acetic acid (DMXAA)-treated Sjögren's syndrome (SS) mouse model. (**A**) Maximum projection of a STED z stack (1 μm) showing TMEM16a (green) and IP$_3$R3 (red) in submandibular gland (SMG) tissue following Huygens deconvolution. The top panel represents the vehicle-treated control, and the bottom panel represents the SS mouse model. Scale bar: 2 μm. Zoomed images highlight the localization of TMEM16a and IP$_3$R3 from the white square on the merged images. (**B**) Diagram illustrating the positioning of apical PM TMEM16a and apical IP$_3$R3 in acinar cells. To analyze the proximity, a 1 μm reference line was drawn across the two parallel TMEM16a over two adjacent acinar cells with IP$_3$R3 aligned vertically in the cytoplasm. (**C–D**) The representative traces of changes in fluorescence of TMEM16a (green) and IP$_3$R3 (red) over the 1 μm distance. (**E**) Analysis of distance between TMEM16a and IP$_3$R3 within cells. (**F**) Analysis of the distance between parallel TMEM16a on adjacent acinar cells. (**G**) Distance measurement of apical IP$_3$R3 between two cells. Each symbol represents the mean of

*Figure 7 continued on next page*

*Figure 7 continued*

5 examinations per image. Vehicle: n=8 replicates from 3 mice; SS mouse model: n=9 replicates from 3 mice. Mean ± SD. Unpaired two-tailed t-test. Source data is included in ***Figure 7—source data 1***.

The online version of this article includes the following source data and figure supplement(s) for figure 7:

**Source data 1.** Raw distance measurements for individual line profiles.

**Figure supplement 1.** No significant alteration in IP$_3$R protein levels in submandibular gland (SMG) in the 5,6-Dimethyl-9-oxo-9H-xanthene-4-acetic acid (DMXAA)-treated mouse model.

**Figure supplement 1—source data 1.** Raw densitometry data.

**Figure supplement 1—source data 2.** Original immunoblots.

as either those derived from genetically modified mice (***Kiripolsky et al., 2017***; ***Yanfei Hu et al., 1992***; ***Shen et al., 2006***; ***Shen et al., 2009***) or those where disease is induced by specific agents or environmental factors. In the context of SS, DMXAA-induced SS can be used to mimic the early stages of the disease which might be triggered in response to bacterial or viral infection. This model is particularly effective in simulating type-1 interferon immune responses seen in early SS, which is thought to contribute to the initial glandular inflammation (***Tabbara and Vera-Cristo, 2000***; ***Papinska et al., 2018***; ***Papinska et al., 2020***). It should also be noted that DMXAA also has been reported to inhibit NAD(P)H quinone oxioreductase (***Phillips, 1999***) and thus the potential increase in free radical load in cells could contribute to the phenotype. The rapid symptom manifestation of disease in the DMXAA-induced model offers an advantage for investigating the early development of SS disease since DMXAA induction is a temporally controlled process, allowing the precise staging of disease onset, thus facilitating studies on the initiating events and ultimately potential early intervention and prevention strategies.

Our studies investigated stimulus-secretion coupling when fluid secretion from SMG and PG in response to physiological stimulation was significantly reduced. Previous work has established a crucial link between an increase in $[Ca^{2+}]_i$ and stimulation of fluid secretion in the salivary glands (***Melvin et al., 2005***; ***Takano et al., 2021***). Efficient secretion is reliant on the specific spatiotemporal regulation of secretagogue-stimulated $[Ca^{2+}]_i$ signals. Given this idea, our initial hypothesis was that a deficiency in secretion after DMXAA administration could be due to reduced or disrupted secretagogue-stimulated $[Ca^{2+}]_i$ signals. Indeed, previous work has revealed that in human SS patient acinar cells and the IL14α knock-in transgenic SS mouse model, CCh-induced $[Ca^{2+}]$ signals were diminished. This reduction was attributed to lower expression levels of the IP$_3$R2 and IP$_3$R3 proteins (***Teos et al., 2015***). To probe this hypothesis, we employed transgenic animals that express the fast $Ca^{2+}$ indicator-GCaMP6f specifically in the acinar cells. Firstly, we validated that the activation of the STING pathway leads to similar salivary gland hypofunction in this genetic background. Surprisingly, DMXAA treatment led to a striking increase in the magnitude of neurally-induced spatially averaged $[Ca^{2+}]_i$ signals. This observation is not consistent with the loss of IP$_3$R proteins being responsible for reduced fluid secretion previously reported in other SS models. Indeed, the expression of IP$_3$R proteins was unchanged following DMXAA treatment. The discrepancy could be attributed to the stage of SS disease represented by the previous studies, with our data presenting an earlier initiating phase of SS disease prior to progression, at a time point before any notable decrease in IP$_3$R proteins has occurred. The molecular mechanism responsible for augmented global $Ca^{2+}$ signals following DMXAA treatment requires further study. Increased $Ca^{2+}$ release/influx, or conversely reduced $Ca^{2+}$ clearance might be responsible. It is tempting to speculate that reduced mitochondrial $Ca^{2+}$ uptake and/or reduced PMCA and SERCA activity as a result of decreased ATP levels may contribute to enhanced cytosolic signals. Nevertheless, we suggest that the augmented $Ca^{2+}$ signals might represent a compensatory mechanism to drive fluid secretion in the face of compromised physiological stimulus-secretion coupling. Although the $Ca^{2+}$ signals were not reduced, the spatiotemporal characteristics of the $Ca^{2+}$ signal were markedly disrupted. Specifically, during neural stimulation, while in control animals there is a pronounced standing gradient of $[Ca^{2+}]$ such that the $[Ca^{2+}]$ is much greater in the apical *vs.* basal aspects of the cell, in DMXAA-treated animals this gradient is largely absent as large changes in $Ca^{2+}$ are propagated to the basal regions of the cells. It is conceivable that the alteration in magnitude coupled with changes in the spatial characteristics of the $Ca^{2+}$ signal contributes to

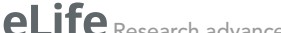

**Figure 8.** Mitochondrial alterations in acinar cells from the 5,6-Dimethyl-9-oxo-9H-xanthene-4-acetic acid (DMXAA)-treated Sjögren's syndrome (SS) mouse model. (**A**) Immunofluorescent staining in submandibular gland (SMG) tissue for ATP5A (green), Na$^+$/K$^+$ ATPase (red), and DAPI for nucleus (blue). The upper panel is the vehicle, and the bottom panel is the SS mouse model. Scale bar: 12 μm. The mitochondrial content was quantified by (**B**) the mitochondrial number per acinar cell and (**C**) the percentage of area occupied by mitochondria per acinar cell. The mitochondrial morphology was analyzed by the (**D**) aspect ratio (AR) for the degree of mitochondrial tubular shape and (**E**) form factor (FF) for the degree of mitochondrial branching (complexity). In (**B**) to (**E**), black dots represent the vehicle condition, and orange squares indicate the SS mouse model. Each symbol represents the mean of 10 cells per image. Vehicle: n=10–15 from 3 mice; SS mouse model: n=10–11 from 3 mice. Mean ± SD. Unpaired two-tailed t-test. Source data is included in *Figure 8—source data 1*.

The online version of this article includes the following source data and figure supplement(s) for figure 8:

**Source data 1.** Raw mitochondrial morphology data.

**Figure supplement 1.** Mitochondrial alterations in the parotid gland of Sjögren's syndrome (SS) mouse model.

**Figure supplement 1—source data 1.** Raw mitochondrial morphology data.

both the defect in fluid secretion and downstream cellular changes including mitochondrial damage to ultimately result in the progression of disease.

We investigated whether changes in the secretory machinery per se were altered in DMXAA-treated animals to result in hyposecretion. Salivary gland fluid secretion is dependent on TMEM16a facilitating Cl$^-$ flux across the apical PM as the driving force for water transport paracellularly and through AGP5 (*Jin et al., 2016*; *Romanenko et al., 2010*). The loss of either TMEM16a or AQP5 results in markedly attenuated fluid secretion (*Romanenko et al., 2010*; *Tsubota et al., 2001*; *Catalán et al., 2015*; *Zeng et al., 2017*). These findings indicate that alteration in expression level, localization, or regulation of these channels could potentially impact fluid secretion. Notably, in DMXAA-treated mice, the AQP5 expression and localization remain unchanged, consistent with a study in human labial

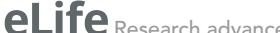

**Figure 9.** Ultrastructural analysis of mitochondria and endoplasmic reticulum (ER) in Sjögren's syndrome (SS) mouse model. (**A-C''**) Images show mitochondrial cristae and ER structure by an electron microscopy (EM) at scales of (**A-A'**) 2 μm, (**B-B'**) 800 nm, and (**C-C'**) 400 nm. (**D**) Mitochondrial perimeter, (**E**) mitochondrial area, and (**F**) circularity were quantified by the shape description in ImageJ. (**G**) Quantification of mitochondrial cristae dispersion was evaluated by the percentage of cristae occupied in one mitochondrion. The (**H**) mean and (**I**) minimum proximity of ER and mitochondria were quantified by the plugin from http://sites.imagej.net/MitoCare/ in ImageJ. Vehicle: n=38 and SS mouse model: n=36 from 3 mice. Mean ± SD. Unpaired two-tailed t-test. Source data is included in *Figure 9—source data 1*.

The online version of this article includes the following source data for figure 9:

*Figure 9 continued on next page*

*Figure 9 continued*

**Source data 1.** Raw mitochondrial morphology data.

minor salivary glands (*Gresz et al., 2015*). We next examined whether the TMEM16a channel function was compromised in the model. Our electrophysiological analysis revealed a significant decrease in TMEM16a activity following CCh-induced stimulation. Again, this reduced activity was not the result of overt mislocalization or lower expression levels of the protein (*Figure 4A and C*). Interestingly, although the secretagogue-stimulated TMEM16a was reduced in acinar cells from DMXAA-treated animals, the sensitivity of the channel to direct activation by $Ca^{2+}$ in the patch pipette appeared unaffected. $IP_3R3$ $Ca^{2+}$ release channels on the ER are located approximately 50–100 nm from TMEM16a on the PM (*Pages et al., 2019*). In this microdomain, confocal microscopy cannot easily distinguish the distinct localization of TMEM16a/$IP_3R$, despite their localization on different membranes. However, STED super-resolution microscopy provides a much higher spatial resolution, achieving 20–80 nm to enable the differentiation of proteins within 20–80 nm of each other. Data using STED microscopy, suggest that the microdomain between apical ER $IP_3R3$ and apical PM TMEM16a is disrupted in the disease model. The severe fragmentation of ER observed in EM images from DMXAA-treated animals also is consistent with an alteration in the relationship between ER and other intracellular domains. The disruption of the relative localization of these channels could conceivably result in diminished TMEM16a activity, if activation is dependent on the local $[Ca^{2+}]$ in its vicinity. Our data showing that the slow $Ca^{2+}$ buffer EGTA eliminates TMEM16a activation in the disease model, but that currents can still be evoked in vehicle-treated controls is consistent with the activation of TMEM16a by the local $Ca^{2+}$ signal surrounding the channel rather than the global cytoplasmic $Ca^{2+}$ signal (*Jin et al., 2016*; *Shah et al., 2020*; *Wang et al., 2020*). Thus, the disruption of this apical microdomain likely alters the local $Ca^{2+}$ signal that TMEM16a experiences leading to reduced activation and fluid secretion.

While changes in cytosolic $[Ca^{2+}]$ are vitally important for stimulating ion flux and hence fluid secretion, $Ca^{2+}$ is also critical for numerous other physiological processes in salivary gland acinar cells. We focused on the potential effects of the dysregulated $Ca^{2+}$ signal on mitochondrial morphology and function. Secretion is an energy-demanding process, necessitating a constant supply of ATP for numerous functions, including vesicle transport, protein modification, membrane fusion, and maintaining ion gradients. For example, the $Na^+/K^+$ ATPase pump generates the $Na^+$ gradient, driving $Cl^-$ transport into the cytosol of acinar cells through NKCC1, and SERCA pumps replenish ER $Ca^{2+}$ levels. In this context, mitochondria are essential as they provide ATP, regulate $Ca^{2+}$ homeostasis, supply metabolic intermediates, and coordinate with the ER to orchestrate cellular functions (*Denton, 2009*; *Detmer and Chan, 2007*). Notably, recent studies have highlighted that mitochondria are abundant and display varied positioning and dynamics in salivary gland cells (*Porat-Shliom et al., 2019*). In SS patients, there are notable alterations in mitochondrial structure, including swelling and disrupted cristae (*Barrera et al., 2021*; *Li et al., 2022*). Correspondingly, mitochondrial-related genes, particularly those involved in metabolism, dynamics, and the electron transport complex, are significantly affected (*Li et al., 2022*). Our data, employing fluorescent immunostaining and EM, mirrors these findings in DMXAA-treated animals. We observed that mitochondrial morphology is altered such that mitochondria are more swollen and rounded, with dispersed cristae, similar to that reported in human SS patients (*Barrera et al., 2021*). Since optimal mitochondrial bioenergetics are also dependent on $Ca^{2+}$ signals, we assessed mitochondrial function by measuring the mitochondrial membrane potential ($\Delta\Psi m$) using a membrane potential sensitive probe and the OCR using Seahorse technology. Our results show that in the SS mouse model, $\Delta\Psi m$, which is critical for ATP synthesis, is diminished (*Figure 10C*). While the ATP-linked OCR remained unchanged, both the basal and maximal OCR were reduced. This suggests that mitochondrial functionality is compromised in the disease model, indicating a decreased capacity to respond to additional cellular stress. An intriguing question arises from these findings: are defects in the function of mitochondria a primary cause of fluid secretion loss in SS, or alternatively is this a consequence of disrupted $[Ca^{2+}]_i$ regulation? Moreover, DNA from damaged mitochondria can activate the cGAS/STING pathway, leading to inflammation (*Decout et al., 2021*; *Gao et al., 2013*). This implies that compromised mitochondria in early SS stages could trigger prolonged inflammation through the STING pathway, potentially contributing to SS progression. Understanding these mechanisms is crucial for developing effective treatments to halt or slow the progression of SS.

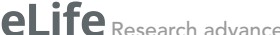

**Figure 10.** Mitochondrial bioenergetics are compromised in the 5,6-Dimethyl-9-oxo-9H-xanthene-4-acetic acid (DMXAA)-treated Sjögren's syndrome (SS) model. (**A**) Mitochondria in the isolated acinar cells were labeled by the MitoTracker Green and co-stained with mitochondrial membrane potential dye, TMRE (red). The merged image shows the colocalization of both dyes, with maximal z-stack projection throughout the acinar cells. (**B**) Representative changes in mitochondrial membrane potential following FCCP-induced depolarization. The vehicle is shown in black; the SS mouse model is in orange. (**C**) The quantification was achieved by the difference of Tetramethylrhodamine, ethyl ester (TMRE) normalized to MitoTracker Green.

*Figure 10 continued on next page*

*Figure 10 continued*

Each dot is the mean of 10 cells from one experiment. Vehicle: n=14 and SS mouse model: n=13 from 3 mice. (**D**) Real-time mitochondrial respiration function was assessed in isolated acinar cells from the vehicle (black) and SS mouse model (orange) using the Seahorse XFe96 extracellular flux analyzer, in response to the pharmacological mito stress (oligomycin, FCCP, rotenone, and antimycin). Vehicle: n=59 and SS mouse model: n=32 from 6 mice. (**E–G**) Mitochondrial respiration function parameters were quantified by oxygen consumption rate (OCR) substracted the non-mitochondrial OCR for (**E**) basal respiration rate, (**F**) ATP-linked respiration rate, and (**G**) maximal respiration rate. Mean ± SD. Unpaired two-tailed t-test. Source data is included in *Figure 10—source data 1*.

The online version of this article includes the following source data and figure supplement(s) for figure 10:

**Source data 1.** Raw mitochondrial bioenrgetics data.

**Figure supplement 1.** Reduction of the basal tetramethylrhodamine, ethyl ester (TMRE) fluorescence in 5,6-Dimethyl-9-oxo-9H-xanthene-4-acetic acid (DMXAA)-treated animals.

**Figure supplement 1—source data 1.** Raw fluorescence data.

# Materials and methods

## Key resources table

| Reagent type (species) or resource | Designation | Source or reference | Identifiers | Additional information |
|---|---|---|---|---|
| Strain, strain background (mouse) | C57BL/6 J | Jackson Laboratory | RRID:IMSR_JAX:000664 | 8–10 weeks-old female mice |
| Strain, strain background (mouse) | B6.129-Bhlha15$^{tm3(cre/ERT2)Skz}$/J | Jackson Laboratory | RRID:IMSR_JAX:029228 | |
| Strain, strain background (mouse) | B6J.Cg-Gt(ROSA)26Sor$^{tm95.1(CAG-GCaMP6f)}$ $^{Hze}$/MwarJ | Jackson Laboratory | RRID:IMSR_JAX:028865 | |
| Chemical compound, drug | 5,6-dimethyl-9-oxo-9H-xanthene-4-acetic acid (DMXAA) | Vadimezan | GC16280 | Stock conc.: 10 mg/ml |
| Chemical compound, drug | Endotoxin 7.5% sodium bicarbonate solution | Sigma-Aldrich | S8761 | Working conc.: 5% |
| Chemical compound, drug | Pilocarpine | Millipore Sigma | P6503 | Stock conc.: 5.63 mg/ml Working conc: 1:100 |
| Antibody | Rabbit polyclonal TMEM16a antibody | Abcam | ab84115 | WB: 1:1000 IHC: 1:250 |
| Antibody | Mouse monoclonal alpha 1 Sodium Potassium ATPase antibody | Abcam | ab2872 | IHC: 1:250 |
| Antibody | Rabbit monoclonal alpha 1 Sodium Potassium ATPase antibody | Abcam | ab76020 | IHC: 1:500 |
| Antibody | Mouse monoclonal ATP5A antibody [15H4C4] | Abcam | ab14748 | IHC: 1:500 |
| Antibody | Rabbit monoclonal Aquaporin 5 antibody | Abcam | ab239904 | WB: 1:1000 IHC: 1:500 |
| Antibody | Rabbit monoclonal STING antibody (D2P2F) | Cell signaling Technology | #13647 | WB: 1:1000 IHC: 1:500 |
| Antibody | Rabbit polyclonal IP$_3$R2 antibody (D2P2F) | PMID:33093175 | | WB: 1:1000 IHC: 1:200 |
| Antibody | Mouse monoclonal IP$_3$R3 antibody (D2P2F) | BD Transduction Laboratory | 610313 | WB: 1:1000 IHC: 1:400 |
| Antibody | Mouse monoclonal Actin antibody | Millipore Sigma | A2228 | WB: 1:10000 |
| Antibody | Donkey anti-rabbit Alexa 488 | ThermoFisher Scientific | A-21206 | IHC: 1:500 |
| Antibody | Donkey anti-mouse Alexa 594 | ThermoFisher Scientific | A-21203 | IHC: 1:500 |
| Antibody | Goat anti-rabbit IgG (H&L) | Invitrogen | SA53557 | WB: 1:10000 |
| Antibody | Goat anti-mouse IgG (H&L) | Invitrogen | SA535521 | WB: 1:10000 |
| Chemical compound, drug | 4',6-Diamidino-2-phenylindol (DAPI) | ThermoFisher Scientific | #62248 | IHC: 1:1000 |
| Chemical compound, drug | MitoTracker Green FM | InvitrogenTM | M7514 | Working conc.: 500 nM |
| Chemical compound, drug | Tetramethylrhodamine, ethyl ester (TMRE) | ThermoFisher Scientific | T669 | Working conc.: 20 nM |
| Chemical compound, drug | Oligomycin | Millipore Sigma | O4876 | Working conc.: 4 µg/ml |
| Chemical compound, drug | Carbonyl cyanide-p-trifluoromethoxyphenylhydrazone (FCCP) | Millipore Sigma | C2920 | Working conc.: 2 µM |
| Chemical compound, drug | Rotenone | Millipore Sigma | R8875 | Working conc.: 2 µM |
| Chemical compound, drug | Antimycin | Millipore Sigma | A8674 | Working conc.: 2 µM |
| Other | Collagenase Type II | Worthington Biochemical | LS004204 | Working conc.: 0.2 mg/ml Reference: https://doi.org/10.1074/jbc.M406201200 |
| Other | Tamoxifen | Sigma-Aldrich | T5648 | Reference: https://doi.org/10.7554/eLife.66170v |
| Software, algorithm | FIJI/Image | Fiji (imagej.net) | | |
| Software, algorithm | Prism | GraphPad | | |

All animal procedures were approved by the University of Rochester Committee on Animal

Resources (UCAR-2001-214E).

## Animals

The murine model of Sjögren's syndrome was established through the induction of the STING pathway (*Bagavant and Deshmukh, 2020*). Briefly, 8–10 weeks old female C57BL/6 J wild-type (WT) mice (Jackson Laboratory; Jax 000664) received subcutaneous injections of DMXAA (Vadimezan; GC16280) at a concentration of 25 mg/kg of body weight on both day 0 and day 21 of the experimental timeline (see *Figure 1*). The control mouse received vehicle (5% sodium bicarbonate; Sigma-Aldrich; S8761), the DMXAA solvent at the corresponding time points. Experiments were performed on day 28 of the experimental timeline. Animals had access to water and were fed ad libitum. All animal procedures were approved by the University of Rochester Committee on Animal Resources (UCAR-2001-214E).

## Evaluation of saliva production

The mice were fasted for 2 hr prior to the evaluation of saliva production. The mice were anesthetized with a solution containing Ketamine (10 mg/mL) and Xylazine (1 mg/ml) by intraperitoneal injection (IP) at a dose of 7 µl/gm body weight over 2 min. The mouse was placed on a heating pad at 37°C during experimentation. A Salimetrics Children's swab (Salimetrics; Cat. no. 5001.05) was placed within the oral cavity of each mouse. The mice were administered the muscarinic agonist pilocarpine (0.375 mg/kg body weight; Millipore Sigma; P6503) by IP injection. Two minutes after the pilocarpine injection, saliva was collected for the following 15 min. The saliva absorbed was subsequently separated from the moist swab through centrifugation at 10,000 rpm for 1 min. The measurement of saliva weight served as a quantitative evaluation of the efficacy of whole saliva secretion. To measure neurotransmitter-stimulated saliva secretion more directly, the mouse was anesthetized as previously described (*Takano et al., 2021*) and a surgical incision was made in the skin to expose the submandibular gland (SMG). The surrounding connective tissue was excised to facilitate positioning within a custom-made 3D-printed gland holder. A pair of stimulation electrodes were attached to the duct bundle and the SMG. The pre-weighed filter paper was positioned within the oral cavity of the mouse to capture saliva secretion. Secretion was initiated by electrical stimulation sequences generated by a stimulus isolator (Iso-flex, A.M.P.I.) set at 5 mA, 200 ms, at frequencies of 1, 3, 5, 7, and 10 Hz with train frequency and duration (typically 1 min) controlled by a train generator (DG2A, Warner Instruments). The interval between each stimulus was 3 min. After stimulation, the filter paper was removed and weighed. The difference between the weight of filter paper before and after the electrode stimulation represented the saliva produced by the respective salivary gland during the given stimulation period.

## In vivo Ca²⁺ imaging

To generate mice expressing a fluorescent $Ca^{2+}$ indicator in exocrine acinar cells, homozygous mice expressing the fast genetically encoded $Ca^{2+}$ indicator GCaMP6f (B6J.Cg-*Gt(ROSA)26Sor*$^{tm95.1(CAG-GCaMP6f)Hze}$/MwarJ) floxed by a STOP cassette, were crossed with heterozygous tamoxifen-inducible Mist1 Cre mice (B6.129-*Bhlha15*$^{tm3(cre/ERT2)Skz}$/J)A week before the DMXAA or 5% sodium bicarbonate injections, tamoxifen (Sigma-Aldrich; T5648) was given to the mice *via* oral gavage at a dose of 0.25 mg/g of body weight for three consecutive days to excise the loxP sites flanking the STOP codon allowing expression of the $Ca^{2+}$ indicator within salivary glands. The mice were anesthetized and gland-exposed, as described previously (*Takano et al., 2021*; *Takano and Yule, 2022*). The immobilized gland was secured within the holder using a cover glass and maintained in Hank's salt solution (HBSS). $Ca^{2+}$ imaging was conducted in vivo via two-photon microscopy using an Olympus FVMPE-RS system equipped with an Insight X3 pulsed laser (Spectra-Physics) utilizing a heated (OKOLab COL2532) 25 x water immersion lens (Olympus XLPlan N 1.05 W MP). GCaMP6f was excited at 950 nm and emission collected between 495–540 nm, with images captured at 0.5 s intervals following stimulation for 10 s with 3 min between stimulation periods. Statistical analyses were performed with two-way ANOVA with multiple comparisons using Prism (GraphPad) as indicated in the figure legends.

## Immunofluorescent staining for sliced tissue

Following verification of decreased saliva secretion in mice, glands were processed for immunocytochemistry. Briefly, the isolated salivary glands were fixed in 4% paraformaldehyde at 4°C overnight. The fixed gland was processed, embedded in paraffin, and subsequently sliced into 5 µm thick

sections. Two temperature-induced antigen retrieval protocols were used either based on HIER buffer (10 mM Tris-base, 1 mM EDTA-dehydrate, pH 9.2) or sodium citrate buffer (10 mM sodium citrate, 0.05% Tween 20, pH 6.0). Gland sections were blocked with the 10% donkey serum in 0.2% PBSA (PBS + BSA) at room temperature (RT) for 1 hr. Sections were incubated with the primary antibody at 4°C overnight (TMEM16a (Millipore Sigma; P6593; 1:250), Na$^+$/K$^+$ ATPase (Abcam; ab2872; 1:250), ATP5A (Abcam; ab14748; 1:500), AQP5 (Abcam; ab239904; 1:500), STING (Cell signaling Technology, Cat. 13647; 1:500)). Following washing, the sections were then incubated with the secondary antibody at RT for 1 hr (Donkey anti-rabbit Alexa 488 (Thermo Fisher Scientific; A-21206; 1:500), Donkey anti-mouse Alexa 594 (Thermo Fisher Scientific; A-21203; 1:500)). Nuclei were identified by incubation in DAPI (Thermo Fisher Scientific; Cat. 62248; 1:1000) at RT for 5 min. Tissue sections were mounted using Immu-Mount solution on a slide and then sealed under a coverslip. Images were acquired by Olympus FV1000MP confocal microscopy employing an Olympus UPlanSApo 60 x oil immersion objective. The analysis of images was performed using FIJI software. Statistical analyses were performed with a t-test using Prism (GraphPad) as indicated in the figure legends.

## Patch clamp electrophysiology

Acinar cells were allowed to adhere to Cell-Tak-coated glass coverslips for 15 min before experimentation. Coverslips were transferred to a chamber containing extracellular bath solution (155 mM tetraethylammonium chloride to block K$^+$ channels, 2 mM CaCl$_2$, 1 mM MgCl$_2$, 10 mM HEPES, pH 7.2). Cl$^-$ currents in individual cells were measured in the whole cell patch clamp configuration using pClamp 9 and an Axopatch 200B amplifier (Molecular Devices). Recordings were sampled at 2 kHz and filtered at 1 kHz. Pipette resistances were 3–5 MΩ, and seal resistances were greater than 1 GΩ. Pipette solutions (pH 7.2) contained 60 mM tetraethylammonium chloride, 90 mM tetraethylammonium glutamate, 10 mM HEPES, 1 mM HEDTA (N-(2-hydroxyethyl) ethylenediamine-N, N', N'-triacetic acid) and 20 µM CaCl$_2$ were used to mimic physiological buffering and basal [Ca$^{2+}$]$_i$ conditions (~100 nM Ca$^{2+}$). Free [Ca$^{2+}$] was estimated using Maxchelator freeware. Agonists were directly perfused onto individual cells using a multibarrel perfusion pipette. The pipette solution for the increased basal [Ca$^{2+}$]$_i$ contained hEDTA and a free [Ca$^{2+}$]$_i$ of 500 nM, 1 mM, or 5 mM to induce calcium-activated Cl$^-$ currents without the addition of any agonists. Experiments comparing EGTA, BAPTA, and HEDTA effects upon chloride currents induced by CCh stimulation contained 5 mM free concentrations of the chelator and 100 nM free [Ca$^{2+}$] in the patch pipette. Chloride currents following agonist application were monitored with a single voltage step to 80 mV from a holding potential of –80 mV every second until current magnitudes reached a plateau. Current-voltage relationships were obtained by 20 mV incremental steps between –80 mv and 120 mV from a holding potential of –50 mV.

## STED microscopy

3D STED microscopy was performed using an Abberior Instruments Expert Line STED microscope equipped with an Olympus UPLSAPO 100x/1.4NA oil immersion objective. Briefly, lobules <1 mm were isolated following injection of saline beneath the capsule with a 29-gauge needle. The connecting tissue was digested in 0.1 mg/ml collagenase containing image buffer at 37°C for 5 min. Then isolated lobules were fixed in 100% methanone at –20°C for 5 min, and subsequently were blocked with 10% BSA in 0.1% PBST (PBS + 0.1% Tween20) at RT for 1 hr with gentle shaking. Isolated lobules were incubated with primary antibodies overnight at 4 °C (TMEM16a (Millipore Sigma; P6593; 1:300), IP$_3$R3 (BD Transduction Laboratory; Cat. 610313; 1:200)). After being washed with 0.1% PBST, the membranes were incubated with secondary antibodies at RT for 1 hr (STAR RED, goat anti-rabbit IgG secondary antibody (Abberior, Cat#STRED-1001–500 UG; 1:1000), Alexa Fluor 594 anti-rabbit IgG secondary antibody (Molecular Probes Cat#A-11037; 1:1000)). The tissue was mounted on the slides with Prolong Gold antifade reagent (Invitrogen; Cat. P36930). Sequential confocal and STED images were obtained following excitation of Alexa Fluor 594 and STAR RED by 594 and 640 nm lasers, respectively. Both fluorophores were depleted in three dimensions with a 775 nm pulsed STED laser. Z-stacks were obtained by collecting images at 50 nm intervals using the 3D STED mode. Rescue STED was employed to minimize the light dosage. Blend mode depth projection images were generated and fluorophore volumes and interfaces between these volumes were analyzed using FIJI.

## Seahorse XF cell mito stress assay

Isolated SMGs were finely minced and subsequently resuspended in a solution composed of 0.5% bovine serum albumin (BSA) in Hank's balanced salt solution (HBSS). To isolate acinar cells, the minced tissue was incubated in 0.5% BSA/HBSS containing 0.2 mg/ml of collagenase type II (Worthington; LS004204) for 30 min. Following this incubation, the suspension of cells was centrifuged at 500 rpm for 1 min and the cellular pellet was then resuspended in 40 µg/ml of Trypsin inhibitor (Millipore; Cat. 65035) to terminate further digestion. The function of mitochondria was assessed in isolated acinar cells by measurement of oxygen consumption rate (OCR) employing a Seahorse XF Cell Mito Stress Test system (Agilent, USA). Briefly, Equal sized SMG cell pellets were suspended in buffer and 10 µl of the acinar cell suspension was seeded into individual wells of Seahorse cell culture microplates coated with 10 uL of Cell-Tak (0.25 mg/ml) and the OCR was determined utilizing the Seahorse XFe96 extracellular flux analyzer following sequential exposure to 4 µg/ml oligomycin (Millipore Sigma; O4876), 4 µM carbonyl cyanide-4 (trifluoromethoxy)phenylhydrazone (FCCP; Millipore Sigma; C2920), and 0.5 µM rotenone/antimycin (Millipore Sigma; R8875; A8674) to measure the quantification of basal respiration, ATP-linked respiration, and maximum respiration rate, respectively. Statistical analyses were performed with a t-test using Prism (GraphPad) as indicated in the figure legends.

## Measurement of mitochondrial membrane potential

Isolated SMG acinar cells were loaded with 20 nM Tetramethylrhodamine, Ethyl Ester (TMRE; ThermoFisher Scientific: T669), and 1 µM of MitoTracker Green (Invitrogen; M7514). Fluorescence of both TMRE and MitoTracker Green was captured simultaneously using an inverted epifluorescence Nikon microscope with a 40 x oil immersion objective. The TMRE fluorescence was excited at 560 nm and emitted light collected at 574 nm; MitoTracker Green was excited at 488 nm and emitted light collected at 530 nm. Images were obtained every 1 s with an exposure of 20 ms and 4 × 4 binning using a digital camera controlled by TILL Photonics, TILLvision software. The acinar cells were exposed to 4 µM FCCP for 3 mins by perfusion to rapidly dissipate the membrane potential. Mitochondrial membrane potential was quantified as the change in the ratio of TMRE/MitoTracker Green fluorescence before and after the administration of FCCP. Statistical analyses were performed with a t-test using Prism (GraphPad) as indicated in the figure legends.

## Western blotting

Finely minced salivary glands were homogenized in a lysis buffer supplemented with protease inhibitor cocktail (Complete mini; Roche Diagnostics) for 16–20 strokes. After incubating on ice for 30 min, solubilized proteins were separated by centrifugation at 13,000 rpm at 4°C for 30 min. 10 µg of protein lysate was loaded on 7.5–12% SDS- polyacrylamide gels. Subsequently, the proteins were transferred to PVDF membranes at a voltage of 35 V at 4 °C overnight. The membrane was blocked with 5% non-fat skimmed milk in TBST (50 mM Tris-HCl, pH 7.5 with 0.1% Tween20) at RT for 1 hr and subsequently incubated with primary antibodies overnight at 4 °C (Actin (Millipore Sigma; A2228; 1:10000), IP$_3$R2 (Antibody Research Corporation; 1:1000), IP$_3$R3 (BD Transduction Laboratory; Cat. 610313; 1:1000), TMEM16a (Abcam; ab84115; 1:1000)). After being washed with 0.1% TBST, the membranes were incubated with secondary antibodies at RT for 1 hr (Goat anti-rabbit IgG (H&L) (Invitrogen; SA535571; 1:10000), Goat anti-mouse IgG (H&L) (Invitrogen; SA535521; 1:10000)). Protein band intensity from western blotting was quantified by FIJI. The relative ratio of DMXAA-treated/ vehicle control was calculated in Excel. Lastly, graphical generation and statistics were performed with a t-test using Prism (GraphPad) as indicated in the figure legends.

## Acknowledgements

The authors gratefully acknowledge the University of Rochester's Center for Advanced Microscopy and Nanoscopy (CALMN) for providing access to Multiphoton microscopy for in vivo live imaging and STED super-resolution microscopy, and for Center for Advanced Research Technologies (CART) for the Electron & cryo Microscopy Resource. We also thank the Flow Cytometry Resource (FCR) for its support with the mitochondrial stress assay. Special thanks to Dr. Paul Brooks for sharing Seahorse Technology XF analyzers and for engaging in discussions for optimization of experiments. Thanks to Dr. Catherine Ovitt for her instruction on tissue staining techniques. Additionally, we wish to express

our appreciation to all members of the Yule laboratory for their invaluable feedback, discussions, and assistance, which have been essential in advancing this study. The work was supported by a grant from NIH (NIDCR) DE014756 (to DIY).

## Additional information

### Funding

| Funder | Grant reference number | Author |
|---|---|---|
| National Institute of Dental and Craniofacial Research | DE014756 | Takahiro Takano |

The funders had no role in study design, data collection and interpretation, or the decision to submit the work for publication.

### Author contributions

Kai-Ting Huang, Investigation, Writing - original draft; Larry E Wagner, Data curation, Investigation; Takahiro Takano, Xiao-Xuan Lin, Harini Bagavant, Investigation; Umesh Deshmukh, Investigation, Methodology; David I Yule, Conceptualization, Formal analysis, Supervision, Funding acquisition, Project administration

### Author ORCIDs

Kai-Ting Huang ⓘ https://orcid.org/0000-0003-3138-3406
Larry E Wagner ⓘ https://orcid.org/0000-0002-5631-8697
David I Yule ⓘ https://orcid.org/0000-0002-6743-0668

### Ethics

All animal procedures were approved by the University of Rochester Committee on Animal Resources (UCAR-2001-214E).

Reviewer #1 (Public review): https://doi.org/10.7554/eLife.97069.3.sa1
Reviewer #2 (Public review): https://doi.org/10.7554/eLife.97069.3.sa2
Reviewer #3 (Public review): https://doi.org/10.7554/eLife.97069.3.sa3
Author response https://doi.org/10.7554/eLife.97069.3.sa4

## Additional files

### Supplementary files

• MDAR checklist

### Data availability

All data generated and analyzed in this study are included in the manuscript and supporting files.

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
