## [Editor Report · eLife assessment]

This manuscript presents **important** observations on the early changes that occur in calcium signaling, TMEM16a channel activation, and mitochondrial dysfunction in salivary gland cells in a murine model of autoimmune Sjögren's disease. The study reports that in response to DMXAA treatment which induces a murine model of Sjögren's disease, salivary gland cells show significant changes in saliva release, calcium signaling, TMEM16a activation, mitochondrial function, and sub-cellular morphology of the endoplasmic reticulum. The work is **compelling** and will be of strong interest to physiologists working on secretion, calcium signaling, and mitochondria.

---

## [Referee Report · Reviewer #1 (Public review)]

Summary:

The authors address cellular mechanisms underlying the early stages of Sjogren's syndrome, using a mouse model in which 5,6-Dimethyl-9-oxo-9H-xanthene-4-acetic acid (DMXAA) is applied to stimulate the interferon gene (STING) pathway. They show that in this model salivary secretion in response to neural stimulation is greatly reduced, even though calcium responses of individual secretory cells was enhanced. They attribute the secretion defect to reduced activation of Ca2+ -activated Cl- channels (TMEM16a), due to an increased distance between Ca2+ release channels (IP3 receptors) and TMEM16a which is expected to reduce the [Ca2+] sensed by TMEM16a. A variety of disruptions in mitochondria were also observed after DMXAA treatment, including reduced abundance, altered morphology, depolarization and reduced oxygen consumption rate. The results of this study shed new light on some of the early events leading to the loss of secretory function in Sjogren's syndrome, at a time before inflammatory responses cause the death of secretory cells.

Strengths:

Two-photon microscopy enabled Ca2+ measurements in the salivary glands of intact animals in response to physiological stimuli (nerve stimulation). This approach has been shown previously by the authors as necessary to preserve the normal spatiotemporal organization of calcium signals that lead to secretion under physiological conditions.

Superresolution (STED) microscopy allowed precise measurements of the spacing of IP3R and TMEM16a and the cell membranes that would otherwise be prevented by the diffraction limit. The measured increase of distance (from 84 to 155 nm) would be expected to reduce [Ca2+] at the TMEM16a channel.

The authors effectively ruled out a variety of alternative explanations for reduced secretion, including changes in AQP5 expression, and TMEM16a expression, localization and Ca2+ sensitivity as indicated by Cl- current in response to defined levels of Ca2+. Suppression of Cl- currents by a fast buffer (BAPTA) but not a slow one (EGTA) supports the idea that increased distance between IP3R and TMEM16A contributes to the secretory defect in DMXAA-treated cells.

Weaknesses:

While the Ca2+ distribution in the cells was less restricted to the apical region in DMXAA-treated cells, it is not clear that this is relevant to the reduced activation of TMEM16a or to pathophysiological changes associated with Sjogren's syndrome.

Despite the decreased level of secretion, Ca2+ signal amplitudes were higher in the treated cells, raising the question of how much this might compensate for the increased distance between IP3R and TMEM16a. The authors assume that the increased separation of IP3R and TMEM16a (and the resulting decrease in local [Ca2+]) outweighed the effect of higher global [Ca2+], but this point was not addressed directly.

The description of mitochondrial changes in abundance, morphology, membrane potential, and oxygen consumption rate were not well integrated into the rest of the paper. While they may be a facet of the multiple effects of STING activation and may occur during Sjogren's syndrome, their possible role in reducing secretion was not examined. As it stands, the mitochondrial results are largely descriptive and more studies are needed to connect them to the secretory deficits in SJogren's syndrome.

---

## [Referee Report · Reviewer #2 (Public review)]

Summary:

This manuscript describes a very eloquent study of disrupted stimulus -secretion coupling in salivary acinar cells in the early stages of an animal model (DMXAA) of Sjogren's syndrome (SS). The study utilizes a range of technically innovative in vivo imaging of Ca signaling, in vivo salivary secretion, patch clamp electrophysiology to assess TMEM16a activity, immunofluorescence and electron microscopy and a range of morphological and functional assays of mitochondrial function. Results show that in mice with DMXAA-induced Sjogren's syndrome, there was a reduced nerve stimulation induced salivary secretion, yet surprisingly the nerve stimulation induced Ca signaling was enhanced. There was also a reduced carbachol (CCh)-induced activation of TMEM16a currents in acinar cells from DMXAA-induced SS mice, whereas the intrinsic Ca-activated TMEM16a currents were unaltered, further supporting that stimulus-secretion coupling was impaired. Consistent with this, high resolution STED microscopy revealed that there was a loss of close physical spatial coupling between IP3Rs and TMEM16a, which may contribute to the impaired stimulus-secretion coupling. Furthermore, the authors show that the mitochondria were both morphologically and functionally impaired, suggesting that bioenergetics may be impaired in salivary acinar cells of DMXAA-induced SS mice.

Strengths:

Overall, this is an outstanding manuscript, that will have a huge impact on the field. The manuscript is beautifully well-written with a very clear narrative. The experiments are technically innovative, very well executed and with a logical design The data are very well presented and appropriately analyzed and interpreted.

Review of Revised Manuscript:

The authors have now addressed all my comments and concerns in the revised manuscript to my satisfaction.

---

## [Referee Report · Reviewer #3 (Public review)]

Summary:

The pathomechanism underlying Sjögren's syndrome (SS) remains elusive. The Authors have studied if altered calcium signaling might be a factor in SS development in a commonly used mouse model. They provide a thorough and straightforward characterization of the salivary gland fluid secretion, cytoplasmic calcium signaling and mitochondrial morphology and respiration. A special strength of the study is the spectacular in vivo imaging, very few if any groups could have succeeded with the studies. The Authors show that the cytoplasmic calcium signaling is upregulated in the SS model and the Ca2+ regulated Cl- channels normally localized and function, still fluid secretion is suppressed. They also find altered localization of the IP3R and speculate about lesser exposure of Cl- channels to high local [Ca2+]. In addition, they describe changes in mitochondrial morphology and function that might also contribute to the attenuated secretory response. Although, the exact contribution of calcium and mitochondria to secretory dysfunction remains to be determined, the results seem to be useful for a range of scientists.

Comments on revised version:

I appreciate the Authors' responses and am satisfied with the revised manuscript.

---

## [Author Response]

The following is the authors’ response to the original reviews.

**Reviewer 1:**
StrengthsWe thank the reviewer for recognizing the strengths of our in vivo Ca2+ measurements, super resolution microscopy and assessment of the secretory dysfunction in the Sjogrens syndrome mouse model.WeaknessesPoint 1: The less restricted Ca2+ signal to the apical region of the acinar cell is not really relevant to the reduced activation of TMEM16a by a local signal at the apical plasma membrane.

We agree that the spatially averaged Ca2+ signal is not indicative of the local Ca2+ signal that activates TMEM16a. The description of the disordered Ca2+ signal in the disease model was intended to simply convey that the Ca2+ signal is altered in the model. Whether or indeed how the altered spatial characteristics of the signal are deleterious is not known but we speculate in the discussion that this contributes to the ultrastructural damage observed.

Point 2. Secretion is decreased but the amplitude of the globally averaged Ca2+ signals are increased. No proof is offered that the greater distance between IP3R and TMEM16a is the reason for decreased secretion in the face of this increased peak signal.

We have now added new data that indicates that the local Ca2+ signal is indeed disrupted in the disease model. We show that in control animals, activation of TMEM16a by application of agonist occurs when the pipette is buffered with the slower buffer EGTA but not with the fast buffer BAPTA In contrast, in cells isolated from DMXAA -treated animals both EGTA and BAPTA abolish the agonist-induced currents (new Figure 6). These data are consistent with our super resolution data showing the distance between IP3R and TMEM16a being greaterand thus presumably is enough to allow buffering of Ca2+ release from IP3R such that it does not effectively activate TMEM16a. These data also would suggest that the increased amplitude of the spatially averaged Ca2+ signal is not sufficient to overcome this structural change.

Point 3. Lack of evidence that the mitochondrial changes are associated with the defect in fluid secretion.

We agree that a causal link between the decreased secretion and altered mitochondrial morphology and function is not established. Nevertheless, we feel it is reasonable to contend that profound changes in mitochondrial morphology observed at the light and EM level, together with changes in mitochondrial membrane potential and oxygen consumption are consistent with contributing to altered fluid secretion given that this is an energetically costly process. We have altered the discussion to reflect these caveats and ideas.

**Reviewer 2:**

We thank the reviewer for their assessment of our work and constructive comments.

**Reviewer 3:**

We thank the reviewer for their careful appraisal of our manuscript and insightful comments.

Point 1: Are all the effects of DMXAA mediated through the STING pathway?

This is an important point because as noted DMXAA has been reported to inhibit NAD(P)H quinone oxireductase that could contribute to the phenotype reported here. In future studies we intend to test other STING pathway agonists such as MSA-2 and perhaps antagonists of the STING pathway. We have added text to the discussion indicating that all the effects observed may not be a result of activation of the STING pathway.

Point 2: As noted, and clarified in the text, the driving force for ATP production is the electrochemical H+ gradient which establishes the mitochondrial membrane potential.

Point 3: The reviewer suggested there was a decrease mitochondrial membrane potential in the absence of a change in TMRE steady state.

We apologize for the confusion generated from the presentation of the figure. We normalized TMRE fluorescence against Mitotraker green fluorescence but as shown, the figure does not reflect that the absolute TMRE fluorescence was indeed decreased. Supplemental figure 4 now shows the basal TMRE fluorescence.

Point 4: Indications that the disruption to ER structure seen in Electron Micrographs contributes to the changes in Ca2+ signal and fluid secretion.

We did not focus on the relative distance between ER and apical PM in the EMs primarily because the ER that projects towards the apical PM is a relatively minor component of the specialized ER expressing IP3R and is difficult to identify. We note that the disruption of the bulk ER as quantitated by altered ER-mitochondrial interfaces and fragmentation is consistent with our super resolution data and thus likely plays a role in the mechanism that results in dysregulated Ca2+ signals and reduced secretion.

**Recommendations to Authors:**

**Reviewing Editor:**
(1) The Editor suggests that we should use the activity of TMEM16a to directly measure the [Ca2+] experienced by the channel.

We now present new additional data. First, we show an extended range of pipette [Ca2+] demonstrating identical Ca2+ sensitivity in DMXAA vs vehicle treated cells (Figure 5). Second, importantly, we now present data evaluating the ability of muscarinic stimulation to activate TMEM16a in the presence of either EGTA (slow Ca2+ buffer) or BAPTA (fast Ca2+ buffer). Notably, currents can be stimulated in control cells when the pipette is buffered with EGTA, but not in DMXAA treated cells. BAPTA inhibits activation in both situations (new Figure 6). These data are consistent with TMEM16a being activated by Ca2+ in a microdomain and that this is disrupted in the disease model.

(2) The Editor asks whether a decrease in IP3R3 in a subset of the samples could account for the decreased fluid secretion.

We think this is unlikely given, as noted by the Editor, that a reduction only occurred in a subset of the samples and statistically there was no significant difference to vehicle-treated animals. Moreover, we would note that there is also no difference in the expression of IP3R2 between experimental groups and in studies of transgenic mice where either IP3R2 or IP3R3 were knocked out individually, there was no effect on salivary fluid secretion, indicating that expression of a single subtype can support stimulus-secretion coupling.

(3) Absolute values for changes in fluorescence (over time) should be included together with SD images.

These have been added in Figure 3.

(4) DMXAA has additional effects to STING activation and thus other STING pathway modulators should be used.

We agree that additional STING agonists should be explored in the future but believe that this is beyond the scope of the present studies. Additional text has been added to the discussion acknowledging the additional targets of DMXAA and that they could contribute to the phenotype.

(5) No causal link between the observed Ca2+ changes and mitochondrial dysfunction.

We agree that no experimental evidence is offered to directly support this contention. Nevertheless, dysregulated Ca2+ signals are well-documented to lead to altered mitochondrial structure and function and thus we feel it not unreasonable to speculate that this is a possibility.

(6) The paper would be improved by directly assessing mechanistic connections between altered Ca2+ signaling and TMEM16a activation.

We agree, please refer to point 1 and new figure 6.

**Reviewer 1:**
(1) Standard Deviation images should be explained and the location of ROI identified.

We contend that Standard Deviation images provide an effective visualization (in a single image) of both the magnitude of the Ca2+ increase and the degree of recruitment of cells in the field of view during the entire period of stimulation. We have added text to describe the utility of this technique. Nevertheless, we now show kinetic traces of the changes in fluorescence over time in both apical and basal regions in Figure 3. We also clarify that the traces shown in Figure 2 are averaged over the entire cell.

(2) The Authors should consider that reduced secretion is because cells are dying.

We believe this is unlikely given the lack of morphological changes in glandular structure and the minor lymphocyte infiltration observed in this model. Nevertheless, we now add data showing that the mass of SMG is not altered in the DMXAA-treated animals compared with vehicle-treated (Figure 1E).

(3) The role of mitochondria in the DMXAA phenotype is unclear. What is the effect of acutely de-energizing mitochondria on fluid secretion.

Since fluid secretion is an energetically expensive undertaking, it is not unreasonable to suggest that compromised mitochondrial function may impact secretion. That being said this could occur at multiple levels- production of ATP to fuel the Na/K pump to establish membrane gradients or to provide energy to sequester Ca2+ among a multitude of targets. This will be a subject of ongoing experiments. We contend that experiments to acutely disrupt salivary mitochondria in vivo while assessing fluid secretion would be difficult experiments to perform and interpret given that local administration of agents to SMG would not effect the other major salivary glands and systemic administration would be predicted to have wide-ranging off target effects.

(4) Could a subset of cells with low IP3R numbers contribute to reduced fluid secretion?

Please see the response to Reviewing Editors point 2.

(5) An attempt to estimate the effect of the spatial distruption of IP3R and TMEM16a localization should be made.

Please see the response to Reviewing Editors point 1.

Minor Points

We have amended the statement form “Highly expressed” to increased.

Regions of the cell have been labelled for orientation in the line scans.

The molecular weight markers have been added in Figure 4.

**Reviewer 2:**
(1) Whether mitochondrial dysfunction is the initiator of the phenotype or a result of the dysregulated Ca2+ signal is unclear.

We agree that our data does not clarify a classic “Chicken vs Egg” conundrum. We plan further experiments to address this issue. Future plans include repeating the mitochondrial and Ca2+ signaling experiments at earlier time points where we know fluid secretion is not yet impacted. This may potentially reveal the temporal sequence of events. Similarly, we plan experiments to mechanistically address why the global Ca2+ signal is augmented- reduced Ca2+ clearance or enhanced Ca2+ release/influx are possibilities. We speculate that reduced Ca2+ clearance, either because mitochondrial Ca2+ uptake is reduced or as a secondary consequence of reduced ATP levels on SERCA and PMCA is a likely possibility.

(2) Measurement of ECAR and direct measurements of ATP and Seahorse methods.

In a separate series of experiments, we monitored ECAR. These data were unfortunately very variable and difficult to interpret, although no obvious compensatory increase was observed. We plan in the future to directly monitor ATP levels in acinar cells using Mg-Green. To normalize for cell numbers in the Seahorse experiments, following centrifugation, cell pellets of equal volume were resuspended in equal volumes of buffer. Acinar cells were seeded onto Cell Tak coated dishes. This information is added to the Methods section.